# Comparative Chromosome Mapping of Musk Ox and the X Chromosome among Some Bovidae Species

**DOI:** 10.3390/genes10110857

**Published:** 2019-10-29

**Authors:** Anastasia A. Proskuryakova, Anastasia I. Kulemzina, Polina L. Perelman, Dmitry V. Yudkin, Natalya A. Lemskaya, Innokentii M. Okhlopkov, Egor V. Kirillin, Marta Farré, Denis M. Larkin, Melody E. Roelke-Parker, Stephen J. O’Brien, Mitchell Bush, Alexander S. Graphodatsky

**Affiliations:** 1Institute of Molecular and Cellular Biology, SB RAS, 630090 Novosibirsk, Russia; zakal@mcb.nsc.ru (A.I.K.); polina.perelman@gmail.com (P.L.P.); lemnat@mcb.nsc.ru (N.A.L.); graf@mcb.nsc.ru (A.S.G.); 2Novosibirsk State University, 630090 Novosibirsk, Russia; 3State Research Center of Virology and Biotechnology “Vector”, Federal Service for Surveillance on Consumer Rights Protection and Human Well-being (FBRI SRC VB “Vector”, Rospotrebnadzor), 630559 Koltsovo, Novosibirsk Region, Russia; yudkin_dv@vector.nsc.ru; 4Institute for Biological Problems of Cryolithozone Siberian Branch of RAS, 677980 Yakutsk, Russia; imo-ibpc@yandex.ru (I.M.O.); e.kir@mail.ru (E.V.K.); 5The Royal Veterinary College, Royal College Street, University of London, London NW1 0TU, UK; mfarrebelmonte@gmail.com (M.F.); dlarkin@rvc.ac.uk (D.M.L.); 6School of Biosciences, University of Kent, Canterbury CT2 7NJ, UK; 7Frederick National Laboratory of Cancer Research, Leidos Biomedical Research, Inc., Frederick, MD 21701, USA; melody.roelke-parker@nih.gov; 8Theodosius Dobzhansky Center for Genome Bioinformatics, Saint-Petersburg State University, Sredniy Av. 41A, 199034 Saint-Petersburg, Russia; lgdchief@gmail.com; 9Oceanographic Center, Nova Southeastern University, Fort Lauderdale 3301 College Ave, Fort Lauderdale, FL 33314, USA; 10The Center for Species Survival, Department of Reproductive Sciences, Smithsonian Conservation Biology Institute, Smithsonian’s National Zoological Park, 1500 Remount Road, Front Royal, VA 22630, USA

**Keywords:** BAC-clones, chromosome painting, Kirk’s Dikdik, musk ox, saola, nilgai bull, gaur

## Abstract

Bovidae, the largest family in Pecora infraorder, are characterized by a striking variability in diploid number of chromosomes between species and among individuals within a species. The bovid X chromosome is also remarkably variable, with several morphological types in the family. Here we built a detailed chromosome map of musk ox (*Ovibos moschatus*), a relic species originating from Pleistocene megafauna, with dromedary and human probes using chromosome painting. We trace chromosomal rearrangements during Bovidae evolution by comparing species already studied by chromosome painting. The musk ox karyotype differs from the ancestral pecoran karyotype by six fusions, one fission, and three inversions. We discuss changes in pecoran ancestral karyotype in the light of new painting data. Variations in the X chromosome structure of four bovid species nilgai bull (*Boselaphus tragocamelus*), saola (*Pseudoryx nghetinhensis*), gaur (*Bos gaurus*), and Kirk’s Dikdik (*Madoqua kirkii*) were further analyzed using 26 cattle BAC-clones. We found the duplication on the X in saola. We show main rearrangements leading to the formation of four types of bovid X: Bovinae type with derived cattle subtype formed by centromere reposition and Antilopinae type with Caprini subtype formed by inversion in XSB1.

## 1. Introduction

Cetartiodactyla is a large mammalian order, including camels, whales, pigs, hippos, and ruminants—the suborder of animals with divided stomach. Bovidae is the most specious family in Ruminantia comprising 143 species with 50 genera [1]. Bovidae is generally subdivided into 2 subfamilies: Bovinae (the bovids: tribes Bovini, Tragelaphini and Boselaphini), represented by nilgai, four-horned antelope, wild cattle, bison, Asian buffalo, African buffalo, and kudu; and Antilopinae (antelopes and caprini: Neotragini, Aepycerotini, Cephalophini, Oreotragini, Hippotragini, Alcelaphini, Caprini, Antilopini, and Reduncini tribes), represented by antelopes, gazelles, goats and their relatives [2]. Bovids include several domesticated species (cattle, goat, and sheep) with high economic significance. Although recent advances have been made in the genomic inference of these species, phylogenetic relationships of species within the family are complex and remain somewhat unresolved [3].

Accumulated cytogenetic data for the Bovidae family allow tracing the trends in evolution of karyotypes of the Bovinae [4,5,6,7] and Antilopinae [7,8,9,10,11] subfamilies. Bovid karyotypes are characterized by tandem and Robertsonian translocations of acrocentric chromosomes, producing wide variation in (2n) chromosome numbers [12,13]. Previously, some 43 bovid species have been studied by comparative chromosome painting, mostly with cattle painting probes [14]. Due to its economic importance, the cattle genome has been widely studied, identifying interchromosome rearrangements between species but resolving few intrachromosomal rearrangements. The use of high-resolution dromedary probes [5] and BAC mapping [9,15] has provided increased precision with respect to syntenic segments orientation relative to centromeres and inversions. Proposed ancestral Bovidae and pecoran karyotypes were imputed based on genomic [16] and cytogenetic data [5,16,17]. Although the diploid chromosome number of Bovidae species ranges from 30 to 60, the number of autosomal arms in karyotypes is stable at 58 for most karyotyped species [18]. Comparative chromosome painting data have been applied to phylogenetic analyses of relationships between tribes. For example, a clear marker association was detected for Tragelaphini: the translocation of cattle chromosomes 1;29 [8], which is shared by all members of this group. Because large chromosomal rearrangements by and large correspond in a parsimony sense to morphology-based phylogenies for the group, it seems that chromosomal rearrangements played an important role in the speciation of the Bovidae family [12].

The X chromosome in the Bovidae family presents a special case in Bovidae evolution. displaying marked karyotypic variability between species, in contrast to highly conserved X chromosome morphology seen for the majority of eutherian mammals [19]. Surprisingly, unlike conserved X chromosome of the majority of eutherian mammals noted by Ohno [19], bovid chromosome X displayed variability. Several types of the X chromosome were identified: the cattle type appeared to be submetacentric, while the tragelaphines and antilopinae was acrocentric [20].

Cytogenetic maps of *Bos taurus*, *Bubalus bubalis*, and *Ovis aries* X chromosomes, when compared with that of *Homo sapiens*, revealed the conservation of pseudoautosomal region position in the Antilopinae subfamily, which was detected using a molecular cytogenetics approach [21,22]. In Ruminants and, especially, in the Bovidae family, a substantial number of intrachromosomal rearrangements, including inversions and centromere repositions, have been identified previously by using comparative bacterial artificial chromosome (BAC) mapping [15]. The cited research, however, described species from only three tribes: Bovini, Hippotragini, and Caprini. Moreover, several independent bovid lineages show the presence of compound sex chromosomes resulting from gonosome and autosome fusions [8,14,20,21,23]. Here, we extend the list of 18 species studied by detailed X chromosome BAC mapping to include species from three tribes: nilgai bull (Boselaphini), saola, gaur (Bovini), and Kirk’s Dikdik (Antilopini) to reveal intrachromosomal rearrangements that occurred on the X chromosome in Bovidae family.

Among species of Bovidae family, musk ox (*Ovibos moschatus*), deserves special attention. This species belongs to a basal branch of the Caprini tribe [3], and along with reindeer, they are the only ungulates of the Arctic to survive the late Pleistocene extinction associated the most recent retreat from glaciation [24]. There are classic cytogenetic [25,26,27] and molecular cytogenetic data on chromosome fusions [14] in musk ox karyotype. Moreover, there is a huge pool of data describing cetariodactyl karyotypes using chromosome painting [14,28], especially using human and dromedary probes [5,29,30]. Here, we used the combination of these human and dromedary painting probes to establish a detailed comparative chromosome map for musk ox to interpret the descent of this species’ genome organization in an evolutionary context.

## 2. Material and Methods

### 2.1. Species

The list of studied species, diploid chromosome number, and the source of cell lines are presented in Table 1. All cell lines belong to the cell culture collection of general biological purpose (No. 0310-2016-0002) of the Institute of Molecular and Cellular Biology (IMCB) of the Siberian Branch, Russian Academy of Sciences (SB RAS).

### 2.2. Chromosome Preparation

Metaphase chromosomes were obtained from fibroblast cell lines. Briefly, cells were incubated at 37 °C in 5% CO2 in medium αMEM (Gibco), supplemented with 15% fetal bovine serum (Gibco), 5% AmnioMAX-II complete (Gibco) and antibiotics (ampicillin 100 μg/mL, penicillin 100 μg/mL, amphotericin B 2.5 μg/mL). Metaphases were obtained by adding colcemid (0.02 mg/L) and ethidium bromide (1.5 mg/mL) to actively dividing culture for 3–4 hours. Hypotonic treatment was performed with 3 mM KCl, 0.7 mM sodium citrate for 20 min at 37 °C and followed by fixation with 3:1 methanol:glacial acetic acid (Carnoy’s) fixative. Metaphase chromosome preparations were made from a suspension of fixed fibroblasts, as described previously [31,32]. G-banding on metaphase chromosomes prior to fluorescence in situ hybridization (FISH) was performed using the standard procedure [33]. Heterochromatin analysis was performed by the Combined Method of Heterogeneous Heterochromatin Detection (CDAG) [34]. AT- and GC-enriched repetitive sequences were detected by DAPI (40-6-diamidino-2-phenylindol) and CMA3 (chromomycin A3) fluorescent dyes following formamide denaturation and renaturation in hot salt solution.

### 2.3. FISH Probes

The protocol for the selection of BAC-clones was reported previously [15]. Briefly, we selected 26 BAC clones highly conserved among Cetartiodactyla from bovine CHORI-240 library using bioinformatic tools. BAC DNA was isolated using the Plasmid DNA Isolation Kit (BioSilica, Novosibirsk, Russia) and amplified with GenomePlex Whole Genome Amplification kit (Sigma-Aldrich Co., St. Louis, MO, USA). Labeling of BAC DNA was performed using GenomePlex WGA Reamplification Kit (Sigma-Aldrich Co., St. Louis, MO, USA) by incorporating biotin-16-dUTP or digoxigenin-dUTP (Roche, Basel, Switzerland). The list of BAC-clones is shown in Table 2. Plasmid containing ribosomal DNA [35] was amplified and labeled as described above. Telomere repeats were synthesized and labeled in non-template PCR using primers (TTAGGG)5 and (CCCTAA)5 [36]. Human and dromedary chromosome-specific probes were described previously [6,32] and were labeled by DOP-PCR [37] with biotin-16-dUTP or digoxigenin-dUTP (Roche, Basel, Switzerland).

### 2.4. FISH Procedure

Dual-color FISH experiments on G-banded metaphase chromosomes were conducted as described by Yang and Graphodatsky [32]. Tripsin-treated chromosomes were immobilized in 0.5% formaldehyde in PBS followed by formamid denaturing and overnight probe hybridization at 40 °C. Digoxigenin-labeled probes were detected using anti-digoxigenin-CyTM3 (Jackson ImmunoResearch Laboratories, Inc., West Grove, PA, whereas biotin-labeled probes were identified with avidin-FITC (Vector Laboratories) and anti-avidin FITC (Vector Laboratories, Inc., Burlingame, CA, USA). Images were captured and processed using VideoTesT 2.0 Image Analysis System (Zenit, St. Petersburg, Russia) and a Baumer Optronics CCD camera mounted on a BX53 microscope (Olympus, Shinjuku, Japan).

## 3. Results

### 3.1. Comparative Chromosome Map of Musk Ox, Dromedary, and Human

The musk ox karyotype includes six submetacentric and 17 acrocentric autosomes and one sex chromosomal pair (2N = 48) (Figure 1). The fundamental number of autosomal arms in musk ox is 58, which in general is characteristic for karyotypes of the Bovidae family [18]. To establish the genome-wide chromosome comparative map of the musk ox, human and dromedary painting probes were used. The chromosome map (Figure 1) and additional comparison with cattle chromosomes (the reference karyotype for ruminants) and with pecoran ancestral karyotype (PAK) are summarized in Table 2. The painting probes from 22 human (HSA) and 35 dromedary (CDR) autosomal paints revealed 51 and 61 conserved segments on the musk ox karyotype, respectively.

We studied the distribution of repeated sequences in the musk ox karyotype using several methods. FISH analysis revealed the localization of telomere repeat and ribosomal DNA sequences. Six nucleolar organizing regions (NORs) with telomeric localization on OMO 1, OMO 2 and OMO 8 chromosome pairs were identified (Figure 1). Telomeric repeats are situated on terminal regions of chromosomes. The CDAG differential staining revealed centromeric and telomeric GC-enriched heterochromatin with prominent AT- and GC-enriched pericentromeric blocks of heterochromatin on acrocentric chromosomes (Figure 2). Smaller blocks of heterochromatin were observed on meta- and submetacentric chromosomes, except chromosome 6, where a large block of heterochromatin was identified. Enlarged telomeric blocks were observed on at least 3 pairs of autosomes, while only two pairs of small acrocentrics appeared to have repeated sequences distributed over the whole chromosome.

### 3.2. Mapping of the X Chromosome in Bovidae

To investigate the order of conserved syntenic segments on X chromosomes in the Bovidae family, 26 BAC-clones were localized using FISH on X chromosomes of four species (nilgai bull, saola, gaur, and Kirk’s Dikdik) in a series of pairwise FISH experiments (Table 3). In all Bovinae species, similar order of BAC-clones was observed. The same order was observed earlier in Antilopinae subfamily, except for the marker inversion in Caprini [15]. In total, comparative analysis of BAC order revealed identical syntenic blocks: X Syntenic Block 1 (13 BACs, XSB1), X Syntenic Block 2 (7 BACs, XSB2), and X Syntenic Block 3 (6 BACs, XSB3) [15]. Two types of chromosome X changes were identified in a course of Bovidae evolution: a centromere reposition, and inversions of an entire syntenic block. Interestingly, a segmental duplication in XSB3 containing CH-108D16 was detected on saola X (Figure 3).

## 4. Discussion

### 4.1. Evolution of Musk Ox and Bovid Karyotypes

Prior G-banding karyotypes of musk ox [25,26] revealed five fusions that formed submetacentric chromosomes described using cattle microdissected chromosomes [14]. Here, a complete high-resolution comparative map for musk ox karyotype was obtained using human and camel chromosome specific probes and compared to cattle karyotype (Table 2). Our results are in agreement with previous publications showing the origin of musk ox submetacentric chromosomes [14].

We show the presence of prominent heterochromatin blocks at centromeric positions in the musk ox karyotype. Many bovid species are characterized by prominent pericentromeric blocks of heterochromatin and their karyotype evolution is marked by frequent occurrence of Robertsonian fusions [2,18]. Several hypotheses point to the role that repetitive sequences may have in driving chromosome evolution in bovids by increasing the occurrence of Robertsonian translocations due to the physical proximity of centromeres of acrocentric chromosomes during meiosis [38]. As shown previously, repetitive sequences were involved in formation of Robertsonian translocations in mice [38]. Therefore, the presence of heterochromatic blocks on acrocentric chromosomes in bovid species may contribute to the high variability of bovid karyotypes, including the occurrence of cytotypes in many species, high frequency of Robertsonian fusions, and autosome to the X chromosome translocations.

Overall, the obtained comparative map indicates that musk ox karyotype is nearly homologous to the pecoran ancestral karyotype [5] (Table 2). The comparison of the ancestral elements of the musk ox with other pecoran species demonstrates the rearrangements that formed its karyotype, but also more events occurring in different lineages (Figure 5). The musk ox karyotype has evolved from PAK through six fusions (CDR 1 + 2/32, CDR 20 + 5/13, CDR 18 + 12/34/12, CDR 26 + 28/15/28/15/4, CDR 33/10 + 24/30, and CDR 4+17), one fission (CDR 11), and three inversions (on HSA 21/3/21, CDR 22/3/22/3, HSA 12pq’/22q’’12pq’/22q’’) (Figure 4). It is characterized by inversions on ancestral elements A2, C2, E, and a split of U.

The musk ox is a representative of subfamily Antilopinae, Caprini tribe. Both comparative linkage and FISH maps showed one major distinction between ovine (Antilopinae, Caprini) and bovine (Bovinae) karyotypes. This difference resulted from a translocation involving segments homologous to BTA 9 and BTA 14 [4,7,39]. However, this association is not observed in musk ox. Presumably, this is determined by the basal position of Ovibovina [40], suggesting that the BTA 9/14 translocation occurred only in Caprina subtribe.

The ancestral form of PAC A2 is similar to pronghorn (AAM) 1q [5]. However, other bovid species and Moschidae (Siberian musk deer) [30] showed an inversion changing the order of homologous segments into HSA 21/3/21 (Figure 4). Therefore, this inversion likely represents a cytogenetic marker for at least Bovidae+Moschidae. The verification of this human syntenic association is needed in Cervidae where the fission of the synteny was shown for *Muntiacus muntjac* [41]. Although Cervidae have been well studied with bovid [42] and muntjac [41] probes, comparison to human probes is still unknown, hindering the deduction of ancestral rearrangements.

The ancestral pecoran synteny PAK C2 represents an interesting case. An inversion in HSA 12pq’/22q’’ occurred independently in different phylogenetic lineages in Pecora (Antilocapridae, Moschidae, Bovina, Caprina) (Figure 4) [16]. Therefore, there is a hot spot of chromosome evolution in the region homologous to HSA12pq’/22q’’ in Pecora. Additional investigation is required to verify if this inversion occurred in the same region, with an in-depth analysis of the DNA sequence surrounding this region needed to elucidate the genomic elements causing repeated rearrangements. Contrary to what has been previously suggested [5], this ancestral PAK chromosome C2 was composed of HSA 12/22, and not of HSA 12/22/12/22, because the outgroup and many species from basal lineages have the HSA 12/22 association (whales, Java mouse deer, giraffe, okapi, saola, hirola) [5,11,30,43]. Similarly, the independent inversions in PAK chromosome E (CDR 22/3/22/3/22/3) in BTA 7 (Bovina), OAR 5, OMO 8 (Caprini) (CDR 22/3/22/3), and giraffe (Giraffidae) [30], while ancestral conditions were retained in Java mouse deer, pronghorn, Siberian musk deer and saola, mark another hot spot of chromosome evolution that requires further study.

During FISH experiments on the localization of CDR 22 on OMO chromosomes, an additional small region of homology on OMO 22 was detected. This region was also detected in other Bovidae species: cow, sheep (Figure 5), and saola [5]. The sequence homologies of HSA5 = BTA20 are confirmed by ENSEMBLE genome browser data, also blast data of alpaca RH markers from the chromosome homologous to CDR22 show homology with BTA20 (unpublished data). The order of conservative segments on BTA7 is HSA19p/5 and CDR22/3/22/3. These data differ from research published previously reporting HSA5/19p/5 and CDR3/22/3/22 [6].

The split of the ancestral element PAK U is shown in musk ox and also in PNG, BTA, OAR, and DHU karyotypes, thus suggesting that this fission is a marker for the Bovidae lineage [5].

On another ancestral chromosome, PAK N1 centromere reposition events occurred independently on homologous chromosomes in several bovid lineages (Pseudoryina, Ovibovina, Alcelaphini) (Figure 5). Further refinement of the ancestral chromosome N1 was achieved (Figure 4), showing that the order of human syntenic regions on the PAK ancestral chromosome N1 is HSA 22’/12’’/4pq and not 12’/22’’/4pq, as was reported earlier [5]. OMO 1p (HSA 22q/12pq’’/4pq) retained the ancestral order of conserved segments N1 but was then tandemly fused with PAK A2 based on the position of the ancestral centromere.

### 4.2. Bovine X Chromosome Evolution

The family Bovidae includes two major branches: Bovinae and Antilopinae [40]. Earlier cytogenetic studies identified three types of morphological diversity of the X chromosome in Bovidae: an antilopinae type (acrocentric), a tragelaphines type (acrocentric), and a cattle type (submetacentric) [20]. Tragelaphines chromosome X was likely formed from the ancestral pecoran X by two inversions. This type is ancestral in Bovinae and presented in nilgai, saola (Figure 6), and domestic river buffalo [44]. The X chromosome in nilgai and saola are marked by several morphological features. The first one is an autosomal translocation onto chromosome X in nilgai karyotype. In Tragelaphini and Boselaphini tribes, independent autosomal translocations were observed [9,23,45,46]. Such rearrangements have an impact on the behavior of chromosomes in meiosis, manifested as a lowering of synapsis in the pseudoautosomal region [45]. The second one is a segmental duplication of sequences homologous to CH-108D16 in saola (PNG) (Table 3). Unfortunately, we cannot determine whether this segmental duplication is characteristic for the entire species due to the lack of information of other individuals. However, the duplicated region contains genes responsible for intrauterine development and may have an adaptive value.

The cattle subtype of the Х chromosome is formed by centromere reposition of the ancestral X chromosome in Bovinae. This type of the X chromosome is presented in cattle, American bison [15], and gaur (Figure 6). Thus, it appears to be characteristic not only of cattle, but of the whole subtribe Bovina. 

The centromere reposition and one inversion resulted in the formation of an acrocentric caprine type of the X chromosome [15]. This type of the X chromosome is retained in Kirk’s Dikdik and sable antelope [15]. We suggest calling this type of bovid the X – Antilopine type. Another inversion, which occurred within the XSB1 in the Caprini lineage, is an apomorphic phylogenetic marker for this tribe [15] and marks the formation of the Caprini subtype of X. The mapping of *Panthalops hodgsonii* X chromosome would assert this type for the whole Caprini tribe.

In general, the X chromosome is highly conserved in eutherians [19], but several different types of chromosome rearrangements on the cetartiodactyl X have been shown [15]. It was suggested that the evolutionary chromosome rearrangements may reduce gene flow by suppressing recombination and contributing to species isolation [47]. However, in ruminant species, several evolutionary breakpoint regions (EBR) on the X chromosome associated with enhancers were described that may change gene expression [16]. Therefore, these rearrangements may have an adaptive value and an evolutionary meaning.

Overall, we can distinguish four types of the X chromosome in Bovidae: Bovinae type with derived cattle subtype; and Antilopinae type with Caprini subtype. The Bovinae type was formed from the ancestral pecoran X by two inversions, whereas the Antilopinae type was formed by inversion and centromere reposition. The Cattle and Caprini subtypes were created by centromere repositions and inversion in XSB1, respectively.

## 5. Conclusions

Detailed comparative maps were obtained for musk ox karyotype and X chromosomes of four bovids: Kirk’s Dikdik, gaur, saola, and nilgai bull. Large structural rearrangements leading to the formation of the karyotype of the musk ox were shown. In general, its karyotype is close to the putative ancestral karyotype of Pecora infraorder. The detailed analysis of the BAC-clones order across four species and published data allowed illustrating chromosomal rearrangements during the formation of four main types of X chromosomes in the Bovidae family. In summary, conservation in BACs order was shown in the Bovinae and Antilopinae subfamilies.

## Figures and Tables

**Figure 1 genes-10-00857-f001:**
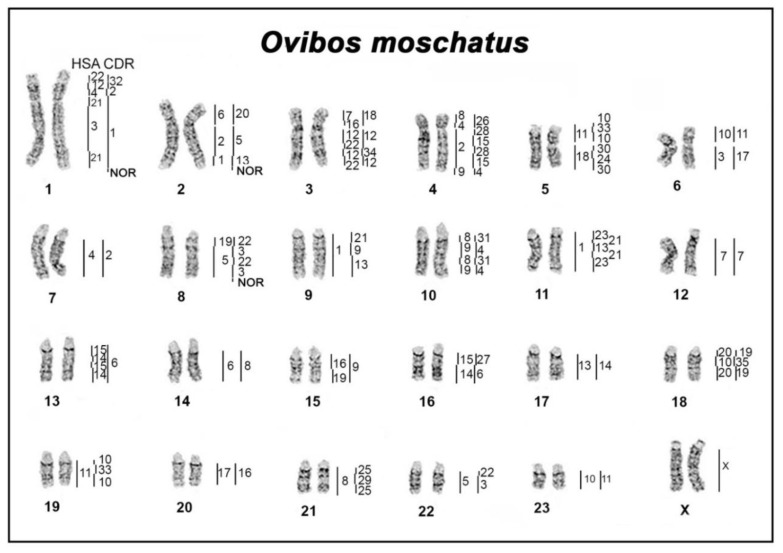
Comparative chromosome painting map of musk ox with homologies to human (HSA) and dromedary (CDR). Nucleolar organizing regions (NOR) show the localization of the nucleolar organizing region.

**Figure 2 genes-10-00857-f002:**
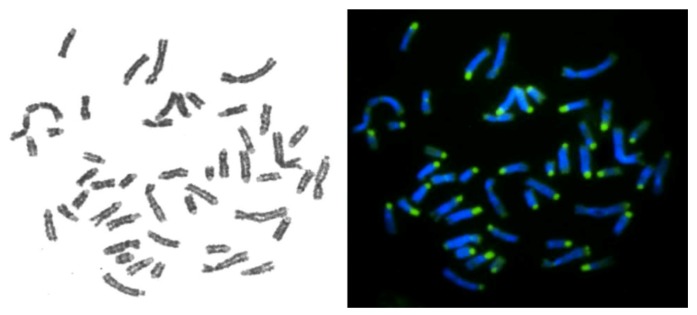
Chromomycin A_3_-DAPI after G-banding (CDAG) staining performed on metaphase chromosomes of musk ox: GTG-banding (**left**) and CMA3/DAPI-staining after denaturation and renaturation procedure (**right**).

**Figure 3 genes-10-00857-f003:**
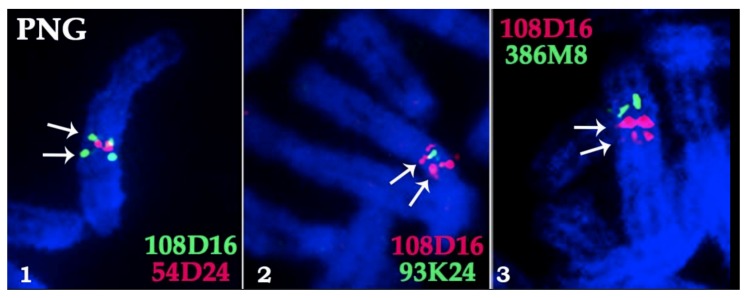
The duplication of the X chromosome segment in *Pseudoryx nghetinhensis* (PNG) shown by dual color FISH of cattle BAC-clones (pink and green) from CHORI-240 library. Three FISH experiments illustrate the revealed order of BACs on the X chromosome: 386M8, 108D16, 54D24, 93K24, 108D16. White arrows indicate the duplicated region corresponding to 108D16: 1, 2 –54D24 and 93K24 are between duplicated regions, 3 –386M8 is outside of duplicated region.

**Figure 4 genes-10-00857-f004:**
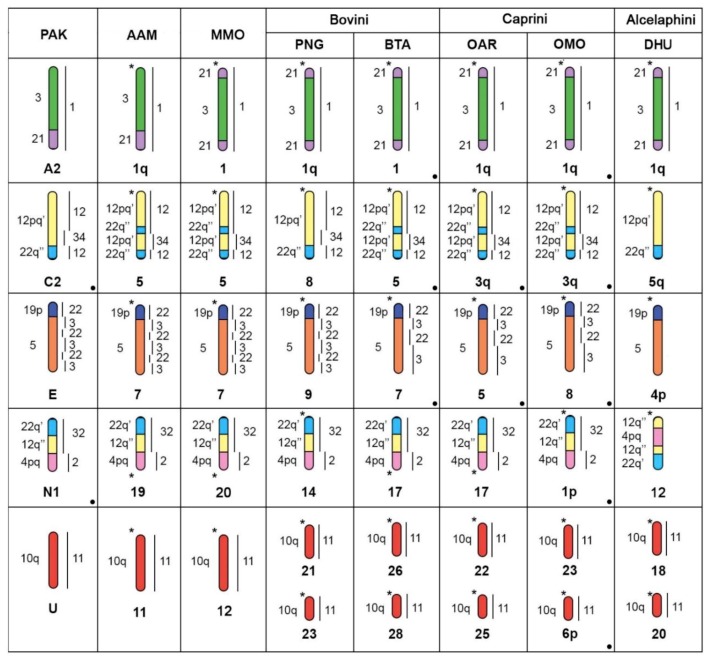
A scheme depicting chromosome homologies of pecoran species to the ancestral karyotype chromosomes (PAK) [5] with human homologies on the left and dromedary on the right. Presented species include AAM (*Antilocapra americana*) [5], MMO (*Moschus moschiferus*) [30], and representatives of different Bovidae tribes: PNG (*Pseudoryx nghetinhensis*) [5], BTA (*Bos taurus*) [6] (Bovini), OAR (*Ovis aries*) [10], OMO (*Ovibos moschatus*) (Caprini), DHU (*Damaliscus hunteri*) [11] (Alcelaphini). Centromere positions are shown by an asterisk. New data obtained in this study are marked by a black circle in cell corners.

**Figure 5 genes-10-00857-f005:**
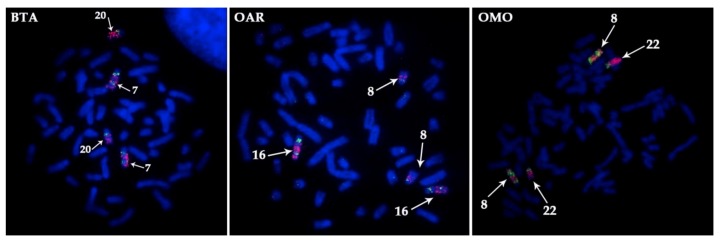
Localization of CDR 22 (green) and CDR 3 (pink) on BTA (*Bos taurus*), OAR (*Ovis aries*), and OMO (*Ovibos moschatus*) metaphase chromosomes by FISH showing additional previously unreported by painting fragment homologous to CDR22. White arrows indicate chromosomes with specific signal.

**Figure 6 genes-10-00857-f006:**
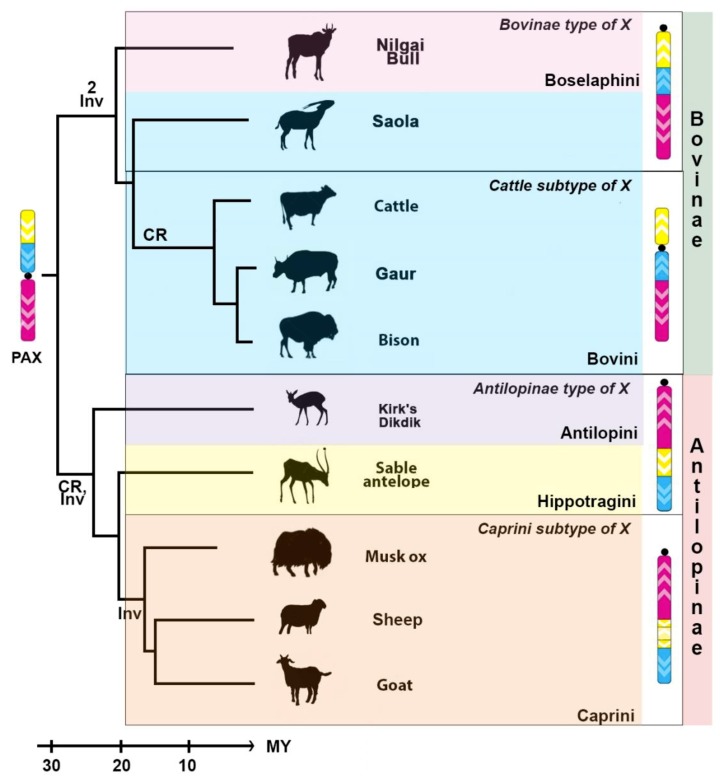
Changes in the structure of the Bovidae X chromosome are depicted on the phylogenetic tree of the family (the tree topology is from [40]). PAX is Pecoran ancestral X chromosome [15]. Major conservative segments are shown in yellow, blue, and pink. Centromere positions are designated by a black circle. White arrowheads show the orientation of the conservative segments. Chromosome changes are shown on phylogenetic tree near respective branches: CR—centromere reposition; and Inv—inversion. Frames show types and subtypes of the bovid X chromosome. The timescale is in million years (MY) of evolution. Nilgai bull X chromosome is shown without autosomal translocation.

**Table 1 genes-10-00857-t001:** The list of Bovidae cell lines used in this study.

Scientific Name, Abbreviation	Common Name	Subfamily	Diploid Number	Sample/Cell Line Source/Acknowledgment
*Ovibos moschatus*, OMO	Musk ox	Antilopinae (Caprini)	48, XX	Allaikhovsky District, Sakha Republic, Yakutia, Russia. IMCB SB RAS, Novosibirsk
*Ovis aries musimon*, OAR	Sheep	Antilopinae (Caprini)	54, XX	Melody Roelke and June Bellizzi, Catoctin Wildlife Preserve and Zoo, Maryland, USA; Laboratory of Genomic Diversity, NCI-Frederick, MD, USA
*Madoqua kirkii*, MKI	Kirk’s Dikdik	Antilopinae (Antilopini)	48, XY	Mitchell Bush, Conservation and Research Center, National Zoological Park, Virginia, USA; Laboratory of Genomic Diversity, NCI-Frederick, MD, USA
*Bos taurus*, BTA	Cattle	Bovinae (Bovini)	60, XX	IMCB SB RAS, Novosibirsk.
*Bos gaurus*, BGA	Gaur	Bovinae (Bovini)	58, XX	Doug Armstrong, Henry Doorly Zoo, Omaha, NE, USA; Laboratory of Genomic Diversity, NCI-Frederick, MD, USA
*Pseudoryx nghetinhensis,* PNG	Saola	Bovinae (Bovini)	50, XX	[5]
*Boselaphus tragocamelus*, BTR	Nilgai bull	Bovinae (Boselaphini)	44, X+14, X+14	Melody Roelke and June Bellizzi, Catoctin Wildlife Preserve and Zoo, Maryland, USA; Laboratory of Genomic Diversity, NCI-Frederick, MD, USA

**Table 2 genes-10-00857-t002:** Correspondence between conserved chromosomal segments in musk ox (OMO), human (HSA), dromedary (CDR), cattle (BTA) and Pecoran ancestral karyotype (PAK) [5] revealed by chromosome painting. The order of conservative segments is started from centromere.

OMO	HSA	CDR	BTA	PAK
1p	22q’/12q”/4pq	32/2	17	N1
1q	21/3/21	1	1	A2
2p	6p	20	23	R
2q	2q”/1	5/13	2	B2
3p	16p/7	18	25	T
3q	12pq’/22q”/12pq’/22q”	12/34/12	5	C2
4p	4/8p”	26	27	V
4q	2pq/9	28/15/28/15/4	11	C1
5p	11	10/33/10	29	W
5q	18	30/24/30	24	S
6p	10q	11	28	U
6q	3	17	22	Q
7	4pq	2	6	F
8	19p/5	22/3/22/3	7	E
9	1	21/9/13	3	A1
10	8p’/9/8p’/9	31/4/31/4	8	B1
11	1	23/21/13/21/23	16	K
12	7	7	4	D
13	15/14/15/14	6	10	G
14	6q	8	9	H1
15	16q/19p	9	18	M
16	15/14	27/6	21	P
17	13	14	12	I
18	20/10p/20	19/35/19	13	J
19	11	10/33/10	15	L
20	17	16	19	N2
21	8q	25/29/25	14	H2
22	5	22/3	20	O
23	10q	11	26	U
X	X	X	X	X

**Table 3 genes-10-00857-t003:** The order of 26 CHORI-240 BACs on Bovidae X chromosomes. The color of the cells corresponds to a given conserved syntenic segment. To display the complete scheme of evolution in Bovidae, the X chromosome maps published previously are also presented [15]. The region duplicated in saola and inverted in Caprini is labeled with a lighter colour.

Syntenic Block	X BAC’s Order in Bovinae Subfamily		X BAC’s Order in Antilopinae Subfamily	
	In Most Bovinae	In Saola	Caprini Tribe	Hippotragini and Antilopini Tribe
X syntenic block 1 (XSB1)	CH240-514O22	CH240-514O22	CH240-66H2	CH240-66H2
	CH240-287O21	CH240-287O21	CH240-155A13	CH240-155A13
	CH240-128C9	CH240-128C9 CH240-106A3	CH240-90L14	CH240-90L14
	CH240-106A3		CH240-373L23	CH240-373L23
	CH240-229I15	CH240-229I15	CH240-62M10	CH240-62M10
	CH240-103E10	CH240-103E10	CH240-122P17	CH240-122P17
	CH240-386M8	CH240-386M8	CH240-252G15	CH240-252G15
X syntenic block 2 (XSB2)	CH240-108D16	CH240-108D16	CH240-375C5	CH240-375C5
	CH240-54D24	CH240-54D24	CH240-130I15	CH240-130I15
	CH240-93K24	CH240-93K24	CH240-118P13	CH240-118P13
		CH240-108D16		
	CH240-122N13	CH240-122N13	CH240-25P8	CH240-25P8
	CH240-195J23	CH240-195J23	CH240-14O10	CH240-14O10
	CH240-316D2	CH240-316D2	CH240-214A3	CH240-214A3
X syntenic block 3 (XSB3)	CH240-214A3	CH240-214A3	CH240-386M8	CH240-386M8
	CH240-14O10	CH240-14O10	CH240-103E10	CH240-103E10
	CH240-25P8	CH240-25P8	CH240-128C9	CH240-229I15
	CH240-118P13	CH240-118P13	CH240-106A3	CH240-106A3
	CH240-130I15	CH240-130I15	CH240-229I15	CH240-128C9
	CH240-375C5	CH240-375C5	CH240-287O21	CH240-287O21
	CH240-252G15	CH240-252G15 CH240-122P17	CH240-514O22	CH240-514O22
	CH240-122P17		CH240-316D2	CH240-316D2
	CH240-62M10	CH240-62M10	CH240-195J23	CH240-195J23
	CH240-373L23	CH240-373L23	CH240-122N13	CH240-122N13
	CH240-90L14	CH240-90L14	CH240-93K2	CH240-93K2
	CH240-155A13	CH240-155A13	CH240-54D24	CH240-54D24
	CH240-66H2	CH240-66H2	CH240-108D16	CH240-108D16

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
