# Peer review of "Comparative Chromosome Mapping of Musk Ox and the X Chromosome among Some Bovidae Species"

_genes, 2019, doi:10.3390/genes10110857_

Round 1

Reviewer 1 Report

This manuscript provides important information for the scientific body of knowledge regarding evolutionary and genetic relationships among Bovidae species. The analyses and interpretations are valid.

I think the article's title should be modified to more specifically reflect the content of the article such as "comparative chromosome mapping of Musk ox and X chromosome among some Bovidae species" etc.

In the body of the text, should species names be italicized?

The English grammar is awkward in a few places that a journal editor should be able to modify. Line 44 whales are not hoofed animals, so need different designation. Several statements about other studies need to be worded better to make it clear that these are results of previous reports not the present study (such as lines 60-62, 79-80). Line 94 should say "pool" not "pull." "data" is a plural concept (lines 183 and 220).

Title of Table 1 should state that these are species studied or available at the specific institution; it is worded to read that this is a global, exhaustive listing (which it is not).

The title of Table 2 is missing cattle (BTA) from title.

Lines 268-270: it is awkward to have a single-sentence paragraph. Also, the way this sentence is worded, it is not obvious what the four X chromosome types are.

Author Response

Response to Reviewer 1

Comments and Suggestions for Authors

This manuscript provides important information for the scientific body of knowledge regarding evolutionary and genetic relationships among Bovidae species. The analyses and interpretations are valid.

Point 1. I think the article's title should be modified to more specifically reflect the content of the article such as "comparative chromosome mapping of Musk ox and X chromosome among some Bovidae species" etc.

Response 1: Thank you so much for this important comment. We changed the title of the paper, as you have suggested.

Point 2. In the body of the text, should species names be italicized?

Response 2: Thank you for this important comment. All Latin names are now in italic.

Point 3. The English grammar is awkward in a few places that a journal editor should be able to modify.

Response 3: Thank you. We put an effort to proofread our manuscript for quality of English.

Point 4. Line 44 whales are not hoofed animals, so need different designation.

Response 4: Thank you. We changed the structure of the sentence.

Line 42-44. Cetartiodactyla is a large mammalian order, including camels, whales, pigs, hippos, and ruminants - suborder of animals with divided stomach. Bovidae is the most specious family in Ruminantia comprising 143 species with 50 genera [1].

Point 5. Several statements about other studies need to be worded better to make it clear that these are results of previous reports not the present study (such as lines 60-62, 79-80).

Response 5: Thank you so much for this comment. We added references and rephrase sentences.

Line 56-62. Previously, some 43 bovid species were studied by comparative chromosome painting, mostly with cattle painting probes [14]. Due to its economic importance, the cattle genome has been widely studied identifying interchromosome rearrangements between species but resolving few intrachromosomal rearrangements. The use of high-resolution dromedary probes [5] and BAC mapping [9,15] provided increased precision about syntenic segments orientation relative to centromeres and inversions.

Line 81-87. In Ruminants and, especially, in the Bovidae family, a substantial number of intrachromosomal rearrangements, including inversions and centromere repositions, were identified previously by using comparative bacterial artificial chromosome (BAC) mapping [15]. The cited research, however, described species from only three tribes: Bovini, Hippotragini, and Caprini. Moreover, several independent bovid lineages show the presence of compound sex chromosomes resulting from gonosome and autosome fusions [8,14,20,21,23].

Point 6. Line 94 should say "pool" not "pull." "data" is a plural concept (lines 183 and 220).

Response 6: Thank you. We corrected it.

Point 7. Title of Table 1 should state that these are species studied or available at the specific institution; it is worded to read that this is a global, exhaustive listing (which it is not).

Response 7: Thank you. We changed the name of the table.

Line 107. Table 1. The list of Bovidae cell lines used in this study

Point 8. The title of Table 2 is missing cattle (BTA) from title.

Response 8: Thank you. We corrected it.

Point 9. Lines 268-270: it is awkward to have a single-sentence paragraph. Also, the way this sentence is worded, it is not obvious what the four X chromosome types are.

Response 9: Thank you so much for this important comment. We improved this concluding paragraph.

Line 315-319. Overall, we can distinguish four types of the X chromosome in Bovidae: Bovinae type with derived cattle subtype and Antilopinae type with Caprini subtype. Bovinae type was formed from the ancestral pecoran X by two inversions whereas Antilopinae type by inversion and centromere reposition. Cattle and Caprini subtypes were created by centromere repositions and inversion in XSB3, respectively.

Reviewer 2 Report

Overall this is an interesting manuscript that adds information to the evolution of the Bovidae family.

Some English language reviews would be good. For example, the use of the word "besides" in the abstract is strange and the authors should either refer to "the X chromosome" or use "chromosome X" but never "X chromosome" without the "the" in front. I would prefer to use the word autosomes, whenever referring to those chromosomes in specific, as they did in the title (but not in other parts of the paper. I am not in a position to provide a full English review, but I recommend the authors doing so.

In the abstract, it is not clear if any comparisons with Bos taurus were performed. Given the commercial relevance of Bos taurus taurus and Bos taurus indicus, I wish the authors would have discussed more these cattle types, in light of their new findings. 

The abstract could include an overall conclusion for the paper.

The methods description is lacking and it relies a lot on previous publications. I would suggest the authors describe a bit more each of the methodologies used. Having a reference is not a substitute for explaining the methods clearly.

Figure 3 is very difficult to see. Figure 2 is also less relevant than a better description of all the rearrangements being observed...

It is lacking from the discussion the impact that rearrangements on the X chromosome, normally very conserved, could have for each species. Any relationship with fertility or reproductive behavior? Or maybe with male/female dimorphism and phenotypes? The discussion could be improved and expanded a little.

Author Response

Response to Reviewer 2

Overall this is an interesting manuscript that adds information to the evolution of the Bovidae family.

Point 1. Some English language reviews would be good. For example, the use of the word "besides" in the abstract is strange and the authors should either refer to "the X chromosome" or use "chromosome X" but never "X chromosome" without the "the" in front. I would prefer to use the word autosomes, whenever referring to those chromosomes in specific, as they did in the title (but not in other parts of the paper. I am not in a position to provide a full English review, but I recommend the authors doing so.

Response 1: Thank you. We put effort to proofread our manuscript for quality of English.

Point 2. In the abstract, it is not clear if any comparisons with Bos taurus were performed. Given the commercial relevance of Bos taurus taurus and Bos taurus indicus, I wish the authors would have discussed more these cattle types, in light of their new findings. 

Response 2: Thank you for this important comment. We didn’t perform direct comparison musk ox with cattle, but we provide the link to cattle chromosomes in table 2. We emphasized in introduction and results section. For the X chromosome mapping we used cattle BACs and made it clear in the abstract.

We did not find any differences in the order of BACs on the X chromosome in the whole subtribe Bovina. We suppose that there are no major differences between the X of cattle subtypes. Although the detailed chromosome mapping or the chromosome level assembly of the Bos taurus indicus is warranted.

Line 32-35. Variations in the X chromosome structure of four bovid species nilgai bull (Boselaphus tragocamelus), saola (Pseudoryx nghetinhensis), gaur (Bos gaurus), and Kirk’s Dikdik (Madoqua kirkii) were further analyzed using 26 cattle BAC-clones.

Line 56-60. Previously, some 43 bovid species were studied by comparative chromosome painting, mostly with cattle painting probes [14]. Due to its economic importance, the cattle genome has been widely studied identifying interchromosome rearrangements between species but resolving few intrachromosomal rearrangements.

Line 152-154. The chromosome map (Figure 1) and additional comparison with cattle chromosomes (the reference karyotype for ruminants) and with pecoran ancestral karyotype (PAK) are summarized in Table 2.

Point 3. The abstract could include an overall conclusion for the paper.

Response 3: Thank you. We add several conclusion sentences.

Line 29-32. We trace chromosomal rearrangements during Bovidae evolution by comparing species already studied by chromosome painting. The musk ox karyotype differs from the ancestral pecoran karyotype by six fusions, one fission, and three inversions. We discuss changes in pecoran ancestral karyotype in the light of new painting data.

Line 35-38. We show main rearrangements leading to the formation of four types of bovid X: Bovinae type with derived cattle subtype formed by centromere reposition and Antilopinae type with Caprini subtype formed by inversion in XSB3.

Point 4. The methods description is lacking and it relies a lot on previous publications. I would suggest the authors describe a bit more each of the methodologies used. Having a reference is not a substitute for explaining the methods clearly.

Response 4: Thank you so much for this important comment. We add more information about methods.

Line 118-122. Heterochromatin analysis was performed by the Combined Method of Heterogeneous Heterochromatin Detection (CDAG) [34]. AT- and GC-enriched repetitive sequences were detected by DAPI (40-6-diamidino-2-phenylindol) and CMA3 (chromomycin A3) fluorescent dyes following formamide denaturation and renaturation in hot salt solution.

Line 126-135. BAC DNA was isolated using the Plasmid DNA Isolation Kit (BioSilica, Novosibirsk, Russia) and amplified with GenomePlex Whole Genome Amplification kit (Sigma-Aldrich Co., St. Louis, MO, USA). Labeling of BAC DNA was performed using GenomePlex WGA Reamplification Kit (Sigma-Aldrich Co., St. Louis, MO, USA) by incorporating biotin-16-dUTP or digoxigenin-dUTP (Roche, Basel, Switzerland). The list of BAC-clones is shown in Table 2. Plasmid, containing ribosomal DNA [35] was amplified and labeled as described above. Telomere repeats were synthesized and labeled in non-template PCR using primers (TTAGGG)5 and (CCCTAA)5 [36]. Human and dromedary chromosome-specific probes were described previously [6,32] and were and labeled by DOP-PCR [37] with biotin-16-dUTP or digoxigenin-dUTP (Roche, Basel, Switzerland).

Line 138-140. Tripsin-treated chromosomes were immobilized in 0.5% formaldehyde in PBS followed by formamid denaturing and overnight probe hybridization at 400C.

Point 5. Figure 3 is very difficult to see. Figure 2 is also less relevant than a better description of all the rearrangements being observed...

Response 5: Thank you. We contrasted Figure 3 and added arrows on duplicated signals. This Figure is very important because such large chromosome segment duplications are detected rarely. Figure 2 contains information about heterochromatin blocks distribution and is important part of the whole description of the musk ox karyotype presented in this paper. We added more descriptions into tResults section and discuss it more in the Discussion section.

Line 165-171. The CDAG differential staining revealed centromeric and telomeric GC-enriched heterochromatin with prominent AT- and GC-enriched pericentromeric blocks of heterochromatin on acrocentric chromosomes (Figure 2). Smaller blocks of heterochromatin were observed on meta- and submetacentric chromosomes, except chromosome 6 were a large block of heterochromatin was identified. Enlarged telomeric blocks were observed on at least 3 pairs of autosomes, while only two pairs of small acrocentrics appeared to have repeated sequences distributed over the whole chromosome.

Line 207-217. We show the presence of prominent heterochromatin blocks at centromeric positions in the musk ox karyotype. Many bovid species are characterized by prominent pericentromeric blocks of heterochromatin and their karyotype evolution is marked by frequent occurrence of Robertsonian fusions [2,18]. Several hypothesis point to the role that repetitive sequences may be driving chromosome evolution in bovids by increasing the occurrence of Robertsonian translocations due to the physical proximity of centromeres of acrocentric chromosomes during meiosis [38]. As shown previously, repetitive sequences were involved in formation of Robertsonian translocations in mice [38]. Therefore, the presence of heterochromatic blocks on acrocentric chromosomes in bovid species may contribute to the high variability of bovid karyotypes, including the occurrence of cytotypes in many species, high frequency of Robertsonian fusions, and autosome to the X chromosome translocations.

Point 6. It is lacking from the discussion the impact that rearrangements on the X chromosome, normally very conserved, could have for each species. Any relationship with fertility or reproductive behavior? Or maybe with male/female dimorphism and phenotypes? The discussion could be improved and expanded a little.

Response 6: Thank you so much for this important comment. There are no precise statements about the interconnection of evolutionary chromosome rearrangements and phenotype. But one recent research shows the correlation of chromosomal rearrangements in evolutionary breakpoint regions and changes in gene expression (Farre et al., 2019). We add several statements to discuss it.

Line 291-293. Such type of rearrangement has an impact on the behavior of chromosomes in meiosis, manifested as lowering of synapsis in the pseudoautosomal region [45].

Line 308-314. In general, the X chromosome is highly conserved in eutherians [19], but several different types of chromosome rearrangements on the cetartiodactyl X have been shown [15]. It was suggested that the evolutionary chromosome rearrangements may reduce gene flow by suppressing recombination and contributing to species isolation [47]. But in ruminant species several evolutionary breakpoint regions (EBR) on the X chromosome associated with enhancers were described that may change gene expression [16]. Therefore these rearrangements may have an adaptive value and an evolutionary meaning.
